# UItron: Foundational GUI Agent with Advanced Perception and Planning

## Abstract

The GUI agent aims to enable automated operations on mobile and PC devices, which is an important task as part of the broader goal of achieving artificial general intelligence. The rapid advancement of visual language models has accelerated the development of GUI agents, owing to their powerful capabilities in visual understanding and task planning. However, building a GUI agent remains a challenging task due to the scarcity of operation trajectories, the lack of interactive infrastructure, and the limitation of initial capabilities in foundation models. In this work, we introduce UItron, an open-source foundational model for automatic GUI agents, featuring advanced GUI perception, grounding, and planning capabilities. UItron highlights the necessity of systemic data engineering and interactive infrastructure as foundational components for advancing GUI agent development. It not only systematically studies a series of data engineering strategies to enhance training effects, but also establishes an interactive environment connecting both Mobile and PC devices. In training, UItron adopts supervised finetuning over perception and planning tasks in various GUI scenarios, and then develops a curriculum reinforcement learning framework to enable complex reasoning and exploration for online environments. As a result, UItron achieves superior performance in benchmarks of GUI perception, grounding, and planning. In particular, UItron highlights the proficiency in interaction with top-tier Chinese mobile Apps, we manually collect over one million steps of operation trajectories across the top 100 most popular Apps, and build offline and online agent evaluation environments. Experimental results demonstrate that UItron achieves significant progress in Chinese App scenarios, propelling GUI agents one step closer to real-world applications.

## 1 Introduction

GUI agents Hong et al. (2024); Zhang & Zhang (2024); Zhang et al. (2024); Yang et al. (2025b); Wu et al. (2025); Gou et al. (2025); Xu et al. (2025); Qin et al. (2025); Lin et al. (2025); Chen et al. (2025) aim to automatically execute complex tasks in various digital environments such as PC and Mobile, satisfying the growing expectations of autonomous decision-making and software control in human-computer interaction. These agents decompose the task instructions into multi-step actions by observing the screen status, then navigate and manipulate the on-screen elements following human-like interactive manners (*i.e.*, click, scroll). This human-like approach provides visually trackable trajectories with a step-by-step task execution process, enabling convenient user interaction and explainable decision-making. Therefore, GUI agents have received a rapidly growing amount of attention, becoming an important research topic toward achieving artificial general intelligence.

The rapid advancement of vision-language models catalyze a series of GUI agents (e.g., Yang et al. (2025b); Wu et al. (2025); Gou et al. (2025); Xu et al. (2025); Qin et al. (2025)) that operate directly on visual GUI images, which achieve superior performance within the framework of unified perception and planning in pure vision. A representative work is UI-TARS Qin et al. (2025), which achieves leading performance via a large amount of data engineering and a carefully designed iterative training framework. Recently, some R1-style works Lu et al. (2025b); Liu et al. (2025); Zhou et al. (2025); Tang et al. (2025); Yang et al. (2025a); Dong et al. (2025) represented by GUI-R1 Zhou et al. (2025) designs multimodal reasoning data and explores the typical group relative policy optimization to improve reasoning ability, reporting improved results in grounding benchmarks. To address the limited adaptability in offline environments, ZeroGUI Yang et al. (2025a) and ARPO Dong et al.

(2025) propose the online reinforcement learning framework, which adopts VLM-based automatic reward estimation to assess task success and continuously learn from GUI environments, without hand-crafted evaluation functions. However, developing automatic GUI agents still remains a highly challenging task due to several limitations: the scarcity of annotated trajectories, the availability of interactive infrastructure, and the sparsity of reinforcement learning reward signals. Training a GUI agent necessitates precise GUI perception, grounding, offline planning, and online planning capabilities, thereby rendering the collection of adequately annotated trajectory data exceedingly difficult. Moreover, in contrast to traditional multimodal tasks, GUI agents face the additional challenge of requiring an interactive environment to execute the model-generated actions that enables multi-round interaction.

In this paper, we introduce UItron, a powerful open-source foundational model for automatic GUI agents, with powerful GUI perception, task grounding, offline/online planning capabilities. UItron emphasizes the importance of data engineering, interactive infrastructure and a progressive curriculum reinforcement learning framework for developing GUI agents. For data engineering, we significantly expand the available operation trajectories through three key aspects: data unification, trajectory distillation, and manual annotation over different domains. Moreover, we systematically investigated a series of data engineering strategies to enhance training effectiveness, including the utilization of various trajectory elements (*e.g.*, observation, thought and action), the exploration of different reasoning formats, and the incorporation of diverse reflection mechanisms like backtracing. We also find the advantages of integrating multi-task UI-related data and general multimodal data. For interactive infrastructure, we build an interactive environment connecting both Mobile and PC devices. It not only simplifies trajectory data collection by automatically recording screenshots and coordinates, but also provides a foundation for online reinforcement learning (RL) during training.

During training, we employ a three-stage strategy across various GUI scenarios, comprising GUI perception, planning, and RL. First, UItron adopts a supervised finetuning training paradigm for GUI perception and planning tasks. The perception task focuses on improving the basic understanding ability of the vision-language model in GUI scenarios, such as grounding, captioning, VQA, and OCR. The planning task focuses on developing the UItron's ability for long-term planning during multi-turn reasoning. Then UItron develops a Curriculum Reinforcement Learning (CuRL) framework with group relative policy optimization algorithm on dynamic trajectory data. To address the issue of sparse rewards in reinforcement learning, CuRL was initially trained in an offline environment (simple), where dense rewards were computed based on action step-level. Subsequently, it was trained in an online environment (complex), where task-level rewards were calculated based on trajectory. Additionally, to boost reward credibility in reinforcement learning, we select trajectories correctly predicted by multiple scoring models through voting.

In particular, UItron emphasizes its ability to interact with top-tier Chinese mobile applications, as we have found that even state-of-the-art solutions typically underperform in these scenarios. To this end, we carefully annotate over one million action steps from the top 100 apps, covering key interaction scenarios like social networking, office tasks, entertainment, and shopping. Based on this, we constructed an offline evaluation dataset to assess the capabilities of GUI agents in Chinese App scenarios, evaluating the performance of different models based on two classical evaluation metrics: single-step success rate and task completion rate. To evaluate the realistic interaction performance of GUI agent in real-world applications, we also build an Android-based cloud real-device environment for online evaluation. Specifically, we develop a rollout method to alternately execute actions and refresh status between GUI agent and Android-based cloud environment. Our experimental results demonstrate that UItron has made substantial progress compared to existing methods in Chinese application scenarios, moving GUI agents toward practical and real-world deployment.

The main contributions of this work are summarized as follows:

- We present a systematic investigation of data engineering and interactive infrastructure that effectively supports the development of foundational GUI agents.

- We develop a curriculum reinforcement learning framework with dense and credible rewards for trajectory data in GUI agents.

- We significantly improve the interactive capabilities of UItron in Chinese scenarios through carefully labeled data and tailored online environments.

- We open-source UItron, achieving superior performance in benchmarks of GUI perception, grounding and offline planning, and competitive results in online agent environments.

## 2 RELATED WORKS

### 2.1 GUI AGENT

With the advent of multimodal large language models (MLLMs), recent GUI agents are primarily based on pure vision approaches. These methods leverage the powerful visual capabilities of MLLMs to process screenshots to understand GUI components. Some researchers have found that scaling data is crucial for improving GUI agent performance, and have proposed using synthetic or video data. For instance, Aria-UI Yang et al. (2025b) and OS-Genesis Yang et al. (2025b) introduce grounding data and trajectory data synthesis pipelines, GUI-explorer Xie et al. (2025) generates function-aware tasks from GUI structure, OS-Atlas Wu et al. (2025) provides a toolkit for multi-platform data synthesis, and GUI-Xplore Sun et al. (2025) enables agents to learn from exploration videos. Other researchers focus on making GUI agents more human-like, such as enabling keyboard and mouse actions Gou et al. (2025), unifying action spaces Xu et al. (2025), supporting cross-application tasks Lu et al. (2025a), and improving interaction and reasoning abilities Qin et al. (2025).

### 2.2 RL WITH AGENT

Recently, researchers have turned to reinforcement learning (RL) to overcome the limitations of supervised finetuning, which relies on manually annotated action trajectories and restricts the autonomous exploration ability of GUI agents. The last work addressed challenges including reasoning patterns Liu et al. (2025); Zhou et al. (2025), sparse rewards Yuan et al. (2025); Tang et al. (2025), and online learning Tang et al. (2025); Wei et al. (2025). For example, InfiGUI-R1 Liu et al. (2025) combines reasoning-augmented trajectories with RL for better error correction, GUI-G1 Zhou et al. (2025) incorporates difficulty-aware objectives, SE-GUI Yuan et al. (2025) and GUI-G2 Tang et al. (2025) propose novel reward formulations to address sparsity, and ZeroGUI Yang et al. (2025a) introduces an online learning framework for automatic task generation and reward estimation. Our work advances this research direction by progressively building the complex task reasoning capability of the model through a two-stage offline-online curriculum reinforcement learning strategy.

## 3 UITRON

In this section, we introduce UItron, an open-source foundational GUI agent framework designed to advance automated interaction and reasoning across both mobile and PC environments. The framework is built upon two key pillars: a robust data engineering pipeline tailored for GUI agent training, and a unified interactive infrastructure that supports scalable data collection and online training. Leveraging these foundations, UItron delivers core capabilities in perception, grounding, and planning, enabling agents to understand complex interfaces, accurately localize UI elements, and perform long-term planning and task execution in complex real-world scenarios.

### 3.1 PROBLEM FORMULATION

GUI agents aim to predict the next action in the $n$-th step based on the task instruction, historical actions and visual environment observation (*i.e.*, screenshot). The action is represented by the action type and additional parameters. Formally, we denote the task instruction as $T$, the historical action as $\{a_1, a_2, ..., a_{n-1}\}$, and the visual environment observation as $o_n$. The task goal of GUI agent in the $n$-th step can be formulated as:

$$a_n = \mathbf{M}_\theta(T, \{a_1, a_2, ..., a_{n-1}\}, o_n), \tag{1}$$

where $\mathbf{M}$ represents the GUI agent with trainable paratemers $\theta$.

### 3.2 DATA ENGINEERING

As shown in Figure 1, we explore systematic data engineering to improve UItron, including perception data, planning data, and distillation data. Besides, we also organize a small amount of general

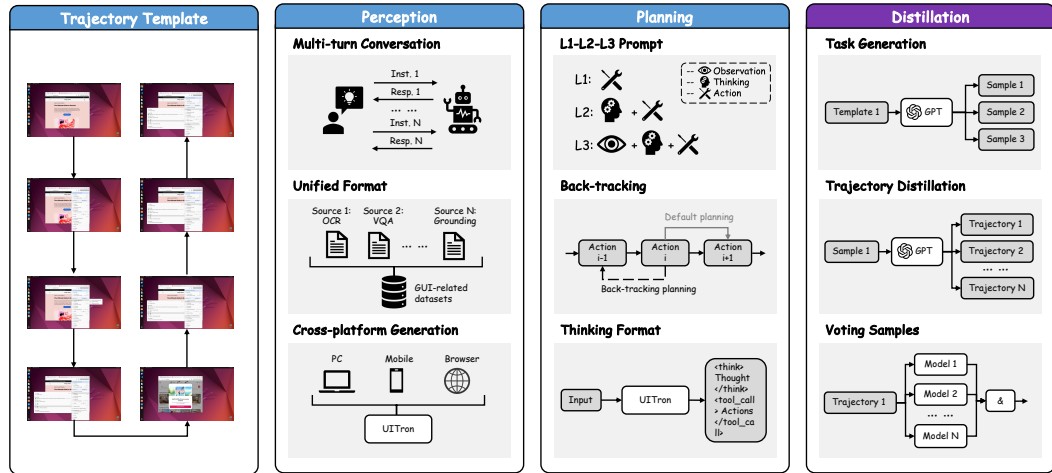

Figure 1: Overall introduction of data engineering.

multimodal data that is beneficial to GUI agent, as well as high-quality manual annotation data for Chinese scenarios.

**Perception Data.** To enhance cross-platform generalization, we unify open-source datasets from diverse sources and synthesis standards, which significantly improves agent localization across different scenarios. Additionally, we reduce training costs and improve UI element comprehension by merging multiple instruction-answer pairs into multi-turn conversations, and we integrate various perception and trajectory data from PC and mobile screenshots into a unified training format.

**Planning Data.** We enhance planning data by introducing backtracking, enabling the model to connect past, present, and future states for more consistent action predictions. Planning data is organized into layers, including screen observation, reasoning, action, and summarization. Backtracking further improves upon standard approaches, which typically predict the next action based only on history and current state, by allowing the model to reflect on how states are reached and thus generate more consistent action predictions.

**Distillation Data.** Manual labeling of long trajectory data is costly, so we developed an automated pipeline that generates tasks using GPT-4o, executes them in simulation, and evaluates results with a VLM-based voting mechanism. This process integrates advanced GUI agent models and tracks the number of attempts for each task, using single-attempt successes for supervised fine-tuning and multiple-attempt cases for GRPO training. Through visual-centric evaluation and strict voting, we ensure high-quality data and categorize sample difficulty to support different training stages, ultimately producing 500k single-step trajectories for training. Details can be found in Appendix A.3.

**General Multimodal Data and Manual Annotation Data.** We enhance GUI agent training by incorporating general multimodal data, such as image-text pairs from OCR, VQA, and image captioning tasks, to improve visual comprehension and adaptability across diverse scenarios. Additionally, to address the lack of Chinese application coverage in existing datasets, we manually annotated operation trajectories from leading Chinese mobile apps, significantly expanding the diversity and representativeness of our training data for both English and Chinese environments.

## 3.3 TRAINING PARADIGM

During training, we employ a three-stage training strategy (as shown in Figure 2), in which consists of two SFT stages for perception and planning tasks, as well as a RL stage with a curriculum reinforcement learning framework. In the first stage, the perception task focuses on improving the basic understanding ability of the VLM in GUI scenarios. In the second stage, the planning task concentrates on predicting the next action based on historical actions. In the final stage, the

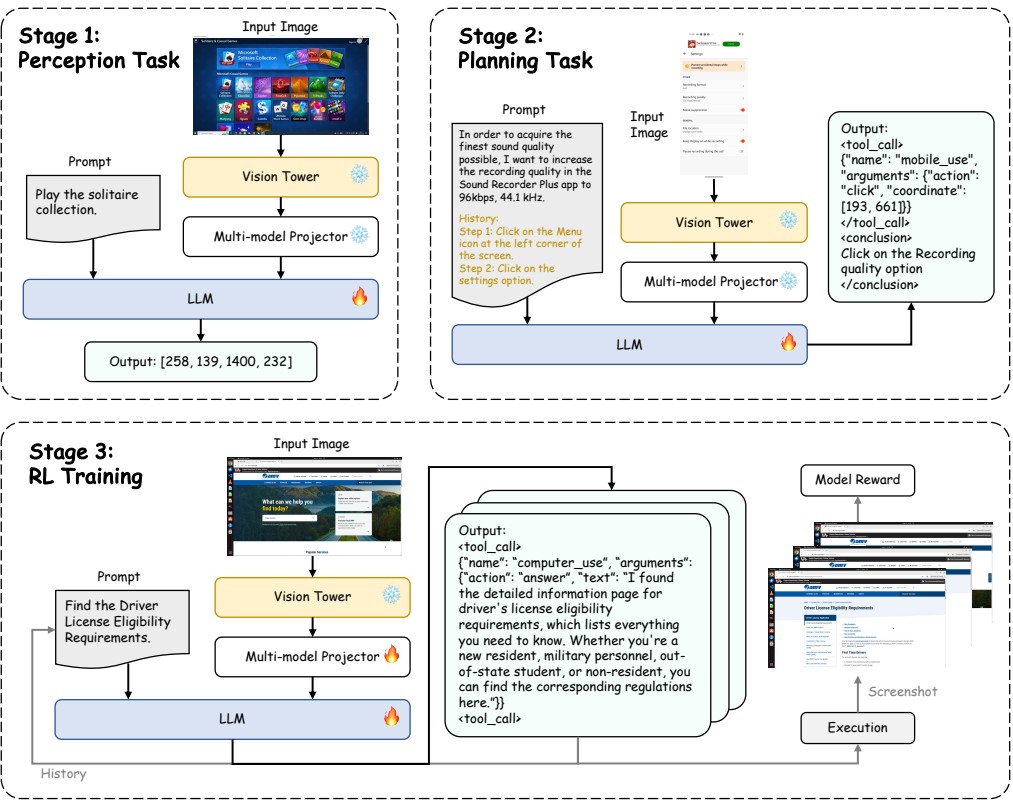

Figure 2: Overall introduction of training paradigm.

curriculum reinforcement learning framework aims to improve reasoning and exploration capacity via group relative policy optimization algorithm on trajectory data.

### 3.3.1 STAGE 1: PERCEPTION TASK

The perceptual abilities of a GUI agent are fundamental for enabling deep understanding and effective interaction with digital interfaces. Without robust perception capabilities, an agent would struggle to interpret the structure, content, and intent behind various UI components.

To address this critical challenge, we initially enhance UItron's perceptual ability in the first stage by fine-tuning it for a wide range of GUI perception scenarios. In particular, we focus on four core perception tasks: grounding, captioning, VQA, and OCR. Grounding enables the agent to accurately localize and associate semantic labels with interface components. Captioning facilitates the generation of descriptions for UI layouts and elements, allowing the agent to summarize and communicate interface structures effectively. Furthermore, VQA empowers the agent to answer queries about the interface by integrating both visual and semantic cues, supporting interactive and context-aware understanding. OCR extracts embedded textual information from the interface, ensuring all relevant data is accessible for downstream reasoning.

Through mastering these perception tasks, UItron attains a holistic and nuanced understanding of user interfaces. It not only enhances its ability to interpret complex environments, but also lays a solid groundwork for advanced reasoning, planning, and autonomous interaction in subsequent stages.

### 3.3.2 STAGE 2: PLANNING TASK

In this stage, training with planning tasks aims to output the predicted actions defined in Equation 2, then optimize it using a generative loss in an auto-regressive manner. Effective planning is a key capability that enables agent to execute purposeful actions and navigate complex digital environments.

The centra idea of planning task is to capture next actions for forward planning and historical actions for backtracking. To this end, we construct two types of training data in the planning task, one for

forward planning and the other for backward backtracking. The forward planning follows the message organization approach defined in Equation (1), which inputs historical actions $(a_1, a_2, ..., a_{n-1})$ and the environment observation $o_n$ to output next action $a_n$. In contrast, the backward backtracking follows the message organization approach defined as follows:

$$a_{n-1}, a_n = \mathbf{M}_\theta(T, (a_1, a_2, ..., a_{n-2}), o_{n-1}, o_n), \tag{2}$$

here the previous action $a_{n-1}$ is not provided in the input, while the agent needs to predict $a_{n-1}$.

### 3.3.3 STAGE 3: CURRICULUM REINFORCEMENT LEARNING

To enhance the reasoning ability, UItron develops a curriculum reinforcement learning framework for performing group relative policy optimization (GRPO) Shao et al. (2024)algorithm on trajectory data. It first computes dense rewards from the action steps in the offline environment (simple), and then computes the task-level reward for the trajectory data in the online environment (complex).

**GRPO.** We adapt the Group Relative Policy Optimization (GRPO) Shao et al. (2024) algorithm for RL. For each input $(x, y)$, the policy $\pi_\theta$ samples a group of $G$ candidate responses $\{o_i\}_{i=1}^G$.

$$\mathcal{J}_{GRPO}(\theta) = \mathbb{E}_{(x,y)\sim\mathcal{D},\{o_i\}_{i=1}^G\sim\pi_{\theta_{old}}(O|x)} \left[ \frac{1}{G} \sum_{i=1}^G \min\left( \frac{\pi_\theta(o_i \mid x)}{\pi_{\theta_{old}}(o_i \mid x)} A_i, \right. \right. \tag{3}$$

$$\left. \left. \text{clip}\left( \frac{\pi_\theta(o_i \mid x)}{\pi_{\theta_{old}}(o_i \mid x)}, 1 - \varepsilon, 1 + \varepsilon \right) A_i \right) - \beta \mathbb{D}_{KL}(\pi_\theta \| \pi_{\text{SFT}}) \right]$$

where $\varepsilon$ and $\beta$ are hyperparameters, and $\pi_{\text{SFT}}$, $\pi_\theta$, and $\pi_{\theta_{old}}$ are the model after SFT, the optimized model and the old policy model. The group-normalized advantage for the $i$-th response is:

$$A_i = \frac{r_i - \text{mean}(\{r_1, r_2, \cdots, r_G\})}{\text{std}(\{r_1, r_2, \cdots, r_G\})} \tag{4}$$

**Offline RL.** Since successful trajectories of GUI agents in online environments are usually rare, the offline RL collects rewards for each action step in the offline environment to avoid sparse rewards. For each action prediction, it generates multiple candidate actions for the same input and calculates the GRPO loss to improve reasoning and exploration capabilities. The training reward integrates assessments of format reward and step-level action accuracy:

$$R = w_f * R_{format} + w_a * R_{action}, \tag{5}$$

where $R_{format}$ denotes if the output conforms to the correct format, $R_{action}$ is calculated by the action type and action parameters:

$$R_{action} = w_t * R_{type} + w_p * R_{param}, \tag{6}$$

where $w_t$ and $w_p$ balance the contribution of each component. $R_{type}$ is assigned to 1 if the predicted action type matches the ground truth, and 0 otherwise. $R_{param}$ assesses the correctness of the action parameter. When the parameter is text, it is 1 only when the predicted content exactly matches ground truth and 0 otherwise. When the parameter is a point, it is assigned a value of 0-1 and decreases as the distance between the predicted coordinates and the actual coordinates increases.

**Online RL.** The online RL is built upon an interactive infrastructure (refer to Appendix A.3 for infrastructure details). It collects rewards for the entire trajectory through the rollout dialogues in an online environment. For each task, online RL allows the model to freely explore all possible action plans until it reaches the maximum number of steps or generates an end signal. To this end, online RL utilizes the advanced vision-language model as a scoring model to evaluate whether the task is completed based on the entire trajectory and task, and outputs a task-level reward signal of 0 or 1.

To improve the credibility of rewards in the RL process, we incorporate multiple scoring models from different vision-language models. Besides, we also strictly filter the trajectories that are predicted correctly by multiple scoring models simultaneously. Finally, the online RL calculates the trajectory-level GRPO loss based on multiple sampled trajectories, thereby improving the exploration ability in the online environment.

| Method | Element Grounding (VWB) | Action Grounding (VWB) | Task Grounding (RefExp) | Element OCR (VWB) | Heading OCR (VWB) | Web QA (VWB) | Web QA (WidgetCap) | Element Caption (WebSRC) |
|---|---|---|---|---|---|---|---|---|
| GPT-4o | 79.91 | 86.41 | – | 79.42 | 64.57 | 77.44 | 61.02 | 78.60 |
| Qwen2.5-VL | 82.81 | 77.67 | 6.55 | 95.72 | 66.82 | 80.23 | 58.45 | 91.2 |
| MutiUI | 75.92 | 36.66 | 43.56 | – | – | – | 72.73 | 82.9 |
| UI-TARS | 96.13 | 92.23 | – | 92.53 | 70.63 | 79.7 | 82.48 | 93.6 |
| UItron-7B | 94.67 | 94.07 | 51.40 | 95.36 | 65.50 | 77.70 | 77.55 | 89.7 |
| UItron-72B | 96.37 | 94.17 | 59.20 | 95.56 | 72.15 | 80.49 | 87.44 | 93.24 |

Table 1: Comparative results on VisualWebBench (VWB) Liu et al. (2024b), RefExp Bai et al. (2021),WidgetCap Li et al. (2020) and WebSRC Chen et al. (2021). Note that GPT-4o and UI-TARS are failed to evaluate due to the invalid output format in task grounding (RefExp).

**Summary.** Finally, we produce two versions named UItron and UItron-RL. The former is obtained after training in stages 1 and 2, while the latter is obtained after reinforcement training in stage 3. In the experiments, we report the results of UItron-RL in all online environments, and report the results of UItron in other offline scenarios.

## 4 EXPERIMENTS

We carry out extensive experiments covering scenarios including GUI perception, grounding, offline planning, and online planning. In particular, we also built our own Chinese scenario evaluation and conduct experiments to explore the improvement of Chinese capabilities.

### 4.1 EVALUATION OF GUI PERCEPTION

As shown in Tables 1, our UItron demonstrates superior performance on perceptual tasks, establishing crucial groundwork for subsequent planning and reasoning capabilities essential for GUI task execution. This effectiveness stems from the limited understanding data employed in both training stages 1 and 2, which not only mitigates the spurious forgetting issue Zheng et al. (2025) that degrades baseline VLLM's original comprehension, but also enhances GUI-specific understanding. This results indicate that maintaining the generalist model's capabilities relevant to the the downstream task while developing specialized skills is critical for creating effective specialist agents.

| Method | Mobile Text | Mobile Icon/Widget | Desktop Text | Desktop Icon/Widget | Web Text | Web Icon/Widget | Avg |
|---|---|---|---|---|---|---|---|
| OS-Atlas-7B | 93.0/95.2 | 72.9/75.8 | 91.8/90.7 | 62.9/63.6 | 90.9/90.6 | 74.3/77.3 | 82.5/84.1 |
| AGUVIS-7B | 95.6/- | 77.7/- | 93.8/- | 67.1/- | 88.3/- | 75.2/- | 84.4/- |
| AGUVIS-72B | 94.5/- | 85.5/- | 95.4/- | 77.9/- | 91.3/- | 85.9/- | 89.2/- |
| UI-TARS-7B | 94.5/96.9 | 85.2/89.1 | 95.9/95.4 | 85.7/85.0 | 90.0/93.6 | 83.5/85.2 | 89.5/91.6 |
| UI-TARS-72B | 94.9/94.8 | 82.5/86.3 | 89.7/91.2 | 88.6/87.9 | 88.7/91.5 | 85.0/87.7 | 88.4/90.3 |
| UItron-7B | 94.1/96.9 | 83.8/88.2 | 94.8/92.8 | 73.6/71.4 | 92.2/91.5 | 81.1/80.8 | 87.7/88.4 |
| UItron-72B | 94.5/95.5 | 88.2/90.5 | 96.9/99.0 | 79.2/80.0 | 93.0/94.0 | 85.4/87.7 | 90.3/92.0 |

Table 2: Performance comparison on ScreenSpot V1 Cheng et al. (2024) and V2 Cheng et al. (2024).

### 4.2 EVALUATION OF GUI GROUNDING

We use ScreenSpot Cheng et al. (2024) and ScreenSpot-V2 Wu et al. (2025) to assess the fundamental GUI-understanding and element-grounding accuracy of GUI-agent models. The experimental results in Table 2 demonstrate that UItron exhibits impressive leading GUI grounding performance across all platforms. This advantage is primarily attributed to UItron's adoption of data engineering specifically tailored for GUI agents, which provides high-quality and well-defined datasets for model training. Furthermore, the parameter scaling experiments of UItron indicate that, with sufficient and high-confidence training data, the model's grounding capability is further enhanced as its scale increases. Compared with state-of-the-art model (*i.e.*, UI-TARS) that additionally utilize internal data, UItron-72B relies solely on open-source data, achieves a 2.1% improvement in micro grounding accuracy.

| Method | AndroidControl-Low[†] | | | AndroidControl-High[†] | | | GUI Odyssey | | | Avg |
|---|---|---|---|---|---|---|---|---|---|---|
| | Type | Grounding | SR | Type | Grounding | SR | Type | Grounding | SR | |
| OS-Atlas | 93.6 | 88.0 | 85.2 | 85.2 | 78.5 | 71.2 | 84.5 | 67.8 | 62.0 | 79.6 |
| AGUVIS | - | - | 80.5 | - | - | 61.5 | - | - | - | - |
| UI-TARS-7B | 98.0 | 89.3 | 90.8 | 83.7 | 80.5 | 72.5 | 94.6 | 90.1 | 87.0 | 87.4 |
| UI-TARS-72B | 98.1 | 89.9 | 91.3 | 85.2 | 81.5 | 74.7 | 95.4 | 91.4 | 88.6 | 88.5 |
| UItron-7B | 96.5 | 90.6 | 90.1 | 91.4 | 77.2 | 79.0 | 95.3 | 85.7 | 84.8 | 87.8 |
| UItron-72B | 96.4 | 93.8 | 92.0 | 93.9 | 85.3 | 86.1 | 94.4 | 86.3 | 86.1 | 90.5 |

Table 3: Comparative results on AndroidControl-Low Li et al. (2024), AndroidControl-High Li et al. (2024) and GUI-Odyssey Lu et al. (2024). [†] indicates that there is no unified evaluation set. Thus we follow UI-TARS Qin et al. (2025) to sample 1000 trajectories for evaluation, ensuring that all test data do not appear in any of the training data.

### 4.3 EVALUATION OF OFFLINE PLANNING

**AndroidControl.** AndroidControl Li et al. (2024) is a benchmark for evaluating the planning and action-execution capabilities of GUI agents on Android devices. As shown in Table 3. UItron-72B achieves the highest grounding and step success rates in both Low and High settings, showcasing exceptional performance not only in guided UI action execution but also in autonomous planning. This underscores the critical importance of our model's unified approach to perception, grounding, and planning, enabling robust and generalizable UI control.

**GUI-Odyssey.** GUI-Odyssey-Random/Task/Device/App are four different test subsets. It aims to assess the generalization ability of autonomous GUI agents across different applications, tasks, and device setups. Notably, UItron remains highly competitive, achieving results close to those of UI-TARS in most metrics. While UItron may perform slightly below UI-TARS on certain cross-app tasks, it consistently demonstrates top-tier results when considering both AndroidControl and GUI-Odyssey benchmarks together, highlighting its overall superiority in comprehensive UI understanding and control. UItron's strong performance across diverse tasks and app combinations underscores its robust generalization ability and reliability, making it one of the most effective agents for complex, real-world UI navigation tasks.

### 4.4 EVALUATION OF ONLINE PLANNING

**OSWorld.** We use OSWorld Xie et al. (2024) to evaluate the performance of GUI agent models as online agents on personal computer (PC) platforms. Table 4 reports the comparative results of UItron and other baseline methods. From the results, we observe that specialized CUA agents generally outperform GUI agents, primarily due to their more singular scenarios and objectives. We can also see that Uitron achieves competitive performance in GUI agents, with only a small gap compared to the state-of-the-art UI-Tars-1.5 method. In addition, the experimental results also show that existing vision-language models such as Qwen25-VL suffers from poor performance, which can be greatly improved through a large amount of targeted training in GUI scenarios.

| Model | Task SR |
|---|---|
| **Compute-Use Agent (CUA)** | |
| OpenAI CUA | 26.0 |
| Claude CUA | 31.2 |
| OpenCUA-32B | 29.7 |
| **GUI Agent** | |
| Augvis-72B | 10.3 |
| UI-TARS-7B | 18.7 |
| UI-TARS-72B | 22.7 |
| UI-TARS-1.5-7B[*] | 23.3 |
| UItron-7B | 23.8 |
| UItron-72B | 26.5 |

Table 4: Task Success Rates (SR) on OSWorld-verified. [*]denotes our reproduction within the same environment.

### 4.5 EVALUATION OF CHINESE SCENARIO

**Evaluation Data.** We evaluate the effectiveness of our method in Chinese environments. To support comprehensive evaluation, we constructed test data and an Android cloud environment. We manually annotate 545 trajectory steps from 109 universal tasks across several apps, and verify that these test tasks did not overlap with the training tasks. Considering that some tasks in the online environment have some app automatic login risks and failures, we retain 86 tasks that can be completed in the online environment.

**Offline Chinese Scenario Results.** Table 5 reports the Step SR and Task SR results of UItron and baseline methods. From the results, we can see that both UItron-7B and UItron-72B significantly outperform the baseline methods in all evaluation metrics, demonstrating their superiority in Chinese scenarios. Interestingly, for the Step SR and Task SR indicators, the results indicate that they have a positive correlation, but the difference in Task SR is significantly larger, which is probably because Task SR reflects the more rigorous accumulation of Step SR. The advanced performance of UItron primarily stems from learning page organization and interaction logic through extensive data from Chinese scenarios, which exhibit significant differences compared to traditional English contexts.

| Method | Step SR | Task SR |
|---|---|---|
| UI-TARS-7B | 75.1 | 22.4 |
| UI-TARS-72B | 80.5 | 32.8 |
| UI-TARS-1.5-7B | 77.4 | 29.3 |
| UItron-7B | 82.7 | 40.5 |
| UItron-72B | 84.1 | 47.4 |

Table 5: Offline evaluation results on top-tier Chinese mobile Apps.

**Online Results.** Table 6 reports the Task SR results of UItron and baseline methods. The results indicate that UItron outperforms the baseline model with a significant performance advantage, verifying its better interaction and exploration capabilities in online environments. Another noteworthy phenomenon is that the online evaluation results of the same model consistently surpass its offline evaluation results, a trend often overlooked in previous research due to the lack of comparable offline and online tasks. The explanation for this phenomenon lies in the nature of the online environment, which offers GUI agents ample space to explore and recover from errors with relaxed constraints. In this setting, certain erroneous steps can be rectified by returning to the original step. Conversely, in offline evaluations, any failed step inevitably results in task failure.

| Method | Task SR |
|---|---|
| UI-TARS-1.5-7B | 38.9 |
| UI-TARS-72B | 49.4 |
| UItron-7B | 56.5 |

Table 6: Online evaluation results on top-tier Chinese mobile Apps.

### 4.6 ABLATION STUDY

To evaluate the effectiveness of the curriculum reinforcement learning framework, we compare models from different training stages: SFT (including Perception Task and Planning Task), Offline RL, and Online RL. The results are shown in Table 7. In the SFT stage, the model is first fine-tuned using collected open-source data and distillation data to establish its foundational capabilities. During the Offline RL stage, step-level rewards are used to enhance the SFT model's single-step actions. Finally, in the Online RL stage, task-level rewards are obtained through rollout dialogues in the online environment to further improve the model. Table 7 reports the effectiveness of our three-stage curriculum reinforcement learning framework.

| Training Stage | Task SR |
|---|---|
| Baseline | 8.83 |
| SFT | 17.7 |
| Offline RL | 21.9 |
| Online RL | 25.2 |

Table 7: Ablations of training Stage. The models are tested on OSworld-verified without Google Drive tasks.

## 5 CONCLUSION

This paper presents UItron, a pioneering open-source foundational model designed to enhance the capabilities of GUI agents in executing complex tasks across digital environments such as PCs and Mobile devices. UItron conduct sufficient investigation of data engineering and interactive infrastructure to handle the scarcity of annotated trajectory data. UItron adopts a typical training paradigm of GUI grounding and planning, and then develops a curriculum reinforcement learning method that improves complex reasoning and exploration in the online environment. In particular, UItron also emphasizes the importance of Chinese interaction capabilities in practical GUI agent deployment. Through comprehensive annotation of over one million action steps from leading Chinese apps, UItron achieves superior results in realistic offline and online evaluation frameworks, bringing GUI agents closer to practical deployment. Experimental results demonstrate that UItron achieves superior performance in benchmarks of GUI perception, task localization and planning, as well as a significant advance in Chinese application scenarios.

## 6    ETHICAL STATEMENT

Our research fully complies with the ethical guidelines for responsible artificial intelligence and machine learning research. This study has undergone and received approval from our institution's internal review process, ensuring that participants' privacy and confidentiality are thoroughly protected. All collected data have been anonymized and are solely used for research purposes.

We recognize that responsible research necessitates careful consideration of potential harms and societal impacts. Our work does not involve manipulative, discriminatory, or unsafe practices, and we have taken extensive measures to minimize any negative impact on participants and the broader community during the design of our experiments.

## 7    REPRODUCIBILITY STATEMENT

We have made every effort to ensure that the results presented in this paper are reproducible. The experimental setup, including training steps and model configurations, is described in detail in the paper. All code is submitted as an attachment for easy copying and verification. The code and model checkpoints will be open-sourced upon paper acceptance.

We believe these measures will enable other researchers to reproduce our work and further advance the field.

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

# A    DATA DETAILS

## A.1    PERCEPTION DATA

**Multi-turn Conversation.**  To minimize redundant image loading and decrease training costs, we consolidated various instruction/description-answer pairs associated with the same screenshot into unified multi-turn conversations, thereby constructing multi-turn training samples. Utilizing such multi-turn data for training not only lowers computational overhead but also improves the model's ability to comprehend and distinguish between different elements within a UI scene.

**Multi-task Unification.** To enhance the basic understanding ability in GUI scenarios, we collect a large amount of UI-related perception data instead of just considering the traditional agentic trajectory data. We collect a wealth of image-text multimodal pairs from a wide range of PC/mobile application screenshots, covering tasks in GUI scenarios such as OCR, VQA, and Caption. We then integrate these UI-related perception data and traditional agentic trajectory data into the unified format to support training.

**Cross-platform Generalization.** To address the generalization challenge in GUI grounding, we integrated data from diverse sources and synthesis methodologies within the GUI agent domain. By unifying open-source datasets (including Uground Gou et al. (2024), Aria-UI Yang et al. (2024), Aguvis Xu et al. (2024) and OS-AtlasWu et al. (2024)), our approach systematically explores whether diverse synthesis criteria can complement one another, thereby enhancing the generalization capability of agent localization across various scenarios.

## A.2    PLANNING DATA

**L1-L2-L3 Inference.** In addition to the final output action, the execution of a planning task can be enhanced by incorporating multiple levels of perception and reasoning to facilitate action prediction. Following  Xu et al. (2025), we divide the planning data into several elements including screen observation, reasoning (thinking), action and summarization.  The L-1 inference involves only action prediction and summarization, L-2 inference further introduces reasoning, and L-3 inference incorporate screen context to observe and analyze changes in the UI interface. This multi-layered and fine-grained perception strategy enables the model to better adapt to tasks of varying complexity and diverse scenarios. To balance efficiency and accuracy, we utilize L2-level descriptions as historical context prompts for action prediction during inference.

**Back-tracking.** The planning process of a GUI agent can be naturally formulated as a partially observable Markov decision process, in which the model predicts the next action based on historical actions and the current state.  However, this approach neglects the model's capacity for reflection and backtracking on previous decisions. Specifically, while the model is aware of its current state, it lacks insight into the sequence of actions that led to that state. Consequently, the model struggles to establish connections between past, present, and future states, which hinders its ability to generate consistent and coherent action predictions. Following Huang et al. (2025), we enhance the interaction between GUI agents and their environment by introducing backtracking.

**Thinking format.** To more precisely distinguish the reasoning process from action prediction during inference optimization, and to facilitate seamless integration with function calls, we employ explicit separators to demarcate different sections of the model output. Specifically, the model's output is structured in the following format:

```
<observation> Observation </observation>
<think> Thought </think>
<tool_call> Actions </tool_call>
<conclusion> Conclusion </conclusion>
```

## A.3    DISTILLATION DATA

Manual labeling of long trajectory data in real scenarios is costly. Therefore, we construct a fully automated trajectory collection process, which includes three stages: (1) Automated task generation

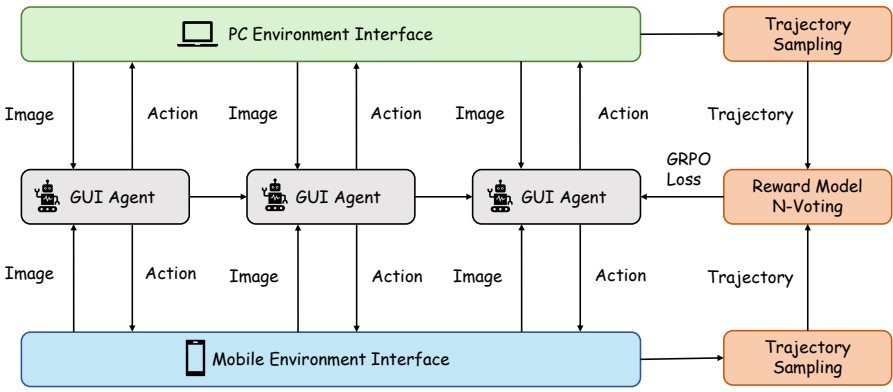

Figure 3: Overall introduction of interactive infrastructure.

guided by real tasks, (2) Automated task execution in simulation environment, and (3) Trajectory result judgment based on VLM voting. After data cleaning and splitting, we finally obtain 500k single-step trajectory data for training.

**Task Generation.** Directly generating tasks with a VLM, without contextual awareness of specific scenarios, often results in unclear or unexecutable tasks. To mitigate this issue, we utilized the initial states of 369 existing tasks in Osworld as prompts to generate additional, related yet distinct tasks using the GPT-4o extension. Furthermore, to prevent task misalignment caused by varying initial states, each generated task is paired with its corresponding initial state.

**Trajectory Distillation.** Building on the multi-domain tasks generated by the VLM, we integrated state-of-the-art GUI agent models into the Osworld simulation environment and implemented a concurrent trajectory distillation pipeline. For each task, the model is allowed up to $n$ attempts, with the reasoning process and specific actions of each step recorded. The complete execution trajectory and task details are then evaluated by a VLM to determine whether the task was successfully completed. Additionally, we tracked the number of attempts for each task: data that succeeded in a single attempt were utilized for supervised fine-tuning (SFT), while data requiring multiple attempts were identified as challenging cases and used for GRPO training.

**Voting Samples.** As illustrated in Figure 1, the trajectory result discriminator is designed according to the following key principles: (1) Visual-Centric Evaluation: GUI agents primarily rely on interactions with the graphical interface. Therefore, changes in the GUI interface serve as the primary indicators of task execution status. (2) Voting Mechanism: Both in supervised fine-tuning (SFT) and GRPO training, even minor variations in training data quality can lead to fluctuations in model prediction accuracy. To ensure robustness, we adopt a stringent voting consensus mechanism, wherein each trajectory is sampled and evaluated multiple times. A trajectory is assigned a positive label only if all evaluations unanimously indicate success. (3) Difficulty Classification: Leveraging the multi-sampling strategy, each task is inferred multiple times by the model. The sample difficulty is then graded based on the number of successful executions across these inferences, enabling targeted application of samples to different training stages.

# B EVALUATION DETAILS

**AndroidControl.** AndroidControl Li et al. (2024) is a benchmark for evaluating the planning and action-execution capabilities of GUI agents on Android devices. It contains 15,283 episodes of everyday tasks across 833 distinct applications, making it the most diverse UI-control dataset to date. Following standard practice, we report results under two settings. AndroidControl-Low: At every step the agent receives a screenshot together with a natural-language description of the required action and must predict both the action type and its exact parameters. AndroidControl-High: Only the high-level task goal and the current screenshot are provided at each step. The agent must autonomously plan the entire procedure and output the correct action together with its parameters. Following OS-Atlas Wu et al. (2025), we reserve 1,000 episodes as an out-of-domain evaluation set and report the action-type accuracy, grounding accuracy, and average step success rate.

| Benchmarks | Platform | #Test Episodes | #Test Samples | History |
|---|---|---|---|---|
| ScreenSpot | Mobile&Desktop&Web | - | 1272 | |
| ScreenSpot-V2 | Mobile&Desktop&Web | - | 1272 | |
| AndroidContorl-Low | Mobile | 1,000 | 6,585 | |
| AndroidContorl-High | Mobile | 1,000 | 6,585 | ✓ |
| GUI-Odyssey-Random | Mobile | 1,933 | 29,426 | ✓ |
| GUI-Odyssey-App | Mobile | 1,139 | 17,455 | ✓ |
| GUI-Odyssey-Device | Mobile | 1,262 | 18,967 | ✓ |
| GUI-Odyssey-Task | Mobile | 1,016 | 17,920 | ✓ |
| OSWorld | Desktop&Web | 369 | - | ✓ |
| AndroidWorld | Mobile | 116 | - | ✓ |
| MobileMiniWob | Web | 92 | - | ✓ |

Table 8: Details of the agentic benchmarks. "Test episodes" refers to the number of trajectory data used for evaluation, while "Test samples" represents the total number of individual step data contained within all trajectories. "History" indicates whether the historical information of previous actions is provided in the model input.

| Benchmarks | Task | Platform | Metric | # Test Samples |
|---|---|---|---|---|
| VisualWebBench | Element Grounding | Web | Prediction Accuracy | 413 |
| | Action Grounding | Web | Prediction Accuracy | 103 |
| | Element OCR | Web | ROUGE-L | 245 |
| | Heading OCR | Web | ROUGE-L | 46 |
| | Web QA | Web | SQuAD-F1 | 314 |
| RefExp | Task Grounding | Web | Accuracy ($IoU \geq 0.5$) | 1000 |
| WidgetCap | Element Caption | Mobile | CIDEr | 1000 |
| WebSRC | WebQA | Web | SQuAD-F1 | 1000 |

Table 9: Details of GUI perception benchmarks. All evaluation data is structured as single-round conversations during experimental.

**GUI-Odyssey.** GUI-Odyssey Lu et al. (2024) is used for evaluating cross-app navigation agents, surpassing the limitation of other benchmarks that are restricted to a single app. It consists of 7,735 episodes, six types of cross-app tasks, 201 apps, and 1.4k app combinations. GUI-Odyssey-Random/Task/Device/App are four different test subsets, with statistics shown in Table 8. It aims to assess the generalization ability of autonomous GUI agents across different applications, tasks, and device setups. Following OS-Atlas Wu et al. (2025), we report the macro average performance across these subsets.

**ScreenSpot.** We use ScreenSpot Cheng et al. (2024) to assess the fundamental GUI-understanding and element-grounding accuracy of GUI-agent models. The ScreenSpot benchmark comprises more than 600 screenshots and 1,200 instructions, spanning multiple platforms—iOS, Android, macOS, Windows, and web pages. We report separate results for Text and Icon/Widget elements on the Mobile, Desktop, and Web splits of ScreenSpot, together with the micro accuracy aggregated across all platforms.

**ScreenSpot-V2.** Similar with ScreenSpot Cheng et al. (2024), we also employ ScreenSpot-V2 Wu et al. (2025) for evaluation, which is a GUI benchmark that advances from basic recognition to cross-modal reasoning. This enhanced version better reflects real-world complexity through optimized annotations, expanded task types, and improved data diversity. The benchmark contains 1,272 instructional samples paired with 756 images, drawing from data sources similar to ScreenSpot.

**VisualWebBench.** We evaluate our model's screen perception capabilities on VisualWebBench Liu et al. (2024b), a comprehensive benchmark containing multiple website-based tasks. For the Grounding Tasks, we measure prediction accuracy by requiring the agent to select correct answers from set of masks (SoM) on screenshots.

**Complicated Perceptual Benchmarks.** To evaluate the model's ability to comprehend abstract instructions, we follow Liu et al. (2024a) by assessing visual grounding of natural language-described elements through RefExp Bai et al. (2021) and testing the reverse task of element captioning on WidgetCap Li et al. (2020). We additionally evaluate on general Web QA task of WebSRC Chen et al. (2021), requiring textual and structural understanding of GUI elements, for further assessing comprehensive perceptual capabilities.

**Baseline Models for Perception Evaluation.** We compare our UItron with SOTA models in both understanding and GUI operation task. Among general VLLMs, we use GPT-4o Hurst et al. (2024) and Qwen2.5-VL Bai et al. (2025) as our baseline for their powerful understanding capabilities in general task understanding; among GUI-related VLLMs, we compare our UItron with MultiUI Liu et al. (2024a) and UI-TARS Qin et al. (2025), the former is specialized in GUI understanding while the later one is the SOTA model in GUI tasks.

**Evaluation Metrics for Chinese Scenario.** We design different evaluation metrics for GUI agents in offline and online environments. For evaluation in offline environment, in which each predicted action corresponding to a ground-truth action, we directly calculate accuracy to evaluate the single-step success rate (*i.e.*, Step SR) and task success rate (*i.e.*, Task SR). A task is deemed successful when all execution steps exactly align with the ground-truth action sequence. For evaluation in online environment, the GUI agent freely explores all possible actions according to finish the task without any ground-truth action. Therefore, we evaluate whether the task is completed accurately based on the complete execution trajectory. We leverage an advanced visual-language model (*i.e.*, GPT-4o Hurst et al. (2024)) to determine whether the task is completed. The result is 1 for completion and 0 for incomplete.

# C    ACTION SPACE

| Environment | Action Space |
|---|---|
| Web | {"name": "computer_use", "arguments": {"action": "key", "keys": ["ctrl", "a"]}}
{"name": "computer_use", "arguments": {"action": "left_click", "coordinate": [x, y]}}
{"name": "computer_use", "arguments": {"action": "type", "text": "text"}}
{"name": "computer_use", "arguments": {"action": "answer", "text": "text"}}
{"name": "computer_use", "arguments": {"action": "terminate", "status": ["success"]}}
{"name": "computer_use", "arguments": {"action": "wait", "time": "time"}}
(Total 15 action types...) |
| Mobile | {"name": "mobile_use", "arguments": {"action": "system_button", "button": "enter"}}
{"name": "mobile_use", "arguments": {"action": "click", "coordinate": [x, y]}}
{"name": "mobile_use", "arguments": {"action": "type", "text": "text"}}
{"name": "mobile_use", "arguments": {"action": "swipe", "coordinate": [x, y], "coordinate2": [x, y]}}
{"name": "mobile_use", "arguments": {"action": "type", "text": "text"}}
{"name": "mobile_use", "arguments": {"action": "status", "button": "success"}}
{"name": "mobile_use", "arguments": {"action": "wait", "time": "time"}}
(Total 11 action types...) |

Table 10: Action space specifications for Web and Mobile environments

# D    LLM USAGE

In the preparation of this manuscript, we employed Large Language Models (LLMs) to assist with writing and refinement. Specifically, the LLM was utilized to enhance the language, readability, and clarity across various sections of the paper. It contributed to tasks such as sentence rephrasing, grammar checking, and improving the overall coherence of the text.

It is crucial to highlight that the LLM played no part in the conceptualization, research methodology, or experimental design. All research ideas, concepts, and analyses were independently developed and executed by the authors. The LLM's role was strictly limited to elevating the linguistic quality of the manuscript, without influencing the scientific content or data analysis.

The authors take full responsibility for the content of the manuscript, including any text generated or refined by the LLM. We have ensured that the contributions made by the LLM comply with ethical standards and do not lead to plagiarism or scientific misconduct.

