# OpenReview forum: "UItron: Foundational GUI Agent with Advanced Perception and Planning"
_ICLR.cc/2026/Conference — ICLR 2026 Conference Withdrawn Submission_

### Official Review · Reviewer_yq78 · 2025-10-23

**Soundness:** 1
**Presentation:** 2
**Contribution:** 2
**Rating:** 2
**Confidence:** 5

**Summary:**

This paper introduces **UItron**, a large-scale foundational GUI agent designed to perform complex tasks across both mobile and desktop environments. The authors identify key challenges in GUI automation—limited high-quality data, lack of scalable interactive infrastructure, and poor generalization—and address them with a unified data engineering pipeline, a dual-platform interactive environment, and a three-stage training paradigm. UItron integrates visual perception, UI element grounding, and multi-step planning into a single model, enabling it to interact with real-world graphical interfaces more effectively.

The proposed training pipeline consists of three progressive stages: (1) **Supervised Fine-Tuning (SFT) for Perception**, focusing on grounding, captioning, VQA, and OCR; (2) **SFT for Planning**, introducing both forward and backward planning data formulations; and (3) **Curriculum Reinforcement Learning (CuRL)** based on the GRPO framework, which leverages both dense offline rewards and sparse online rewards collected through real environment interaction. The system also includes a robust data distillation process using large vision-language models (VLMs) and a voting mechanism to ensure data quality.

**Strengths:**

The paper presents a comprehensive and systematic framework for building foundational GUI agents by combining large-scale data engineering, interactive infrastructure, and curriculum-based reinforcement learning. This integration across perception, grounding, and planning tasks represents a novel formulation compared to prior GUI automation studies that focused on isolated components. The authors design a three-stage training pipeline (Perception SFT → Planning SFT → Curriculum RL) supported by a scalable data distillation process and interactive environments across both mobile and desktop platforms.

**Weaknesses:**

1. Missing details in the training process. The paper proposes a training framework but lacks many crucial implementation details, which raises concerns about reproducibility and the validity of its claims.

- The paper only provides the construction method and scale of the *Distillation Data*, but omits detailed information about the *Perception Data*, *Planning Data*, and *General Multimodal and Manual Annotation Data*. Even if the datasets cannot be released, the authors should provide detailed statistics on data size, data sources, and preprocessing methods.

- Although the paper presents the formula for computing the advantage, it does not clearly define what \(A\) and \(r\) represent. Based on Equation (4), the advantage seems to be computed at the step level. It remains unclear how the *online RL* process converts the *task-level reward signal of 0 or 1* into per-step scores.

- The paper mentions using weighted components in the dense reward function but does not specify the exact weight values or whether any robustness experiments were performed to verify these choices. Similarly, the hyperparameters within the GRPO algorithm are not reported.

- The statement “To improve the credibility of rewards in the RL process, ... by multiple scoring models simultaneously” is underspecified. The paper does not detail which scoring models are included, how they evaluate trajectories, or how their outputs are aggregated into final reward signals.

- Due to the missing RL-related details, the paper should include key diagnostic curves or metrics—such as training/test reward trajectories—to support the credibility of the reported RL training process.

2. Questions regarding the test datasets. The paper uses *AndroidControl* and *GUI-Odyssey* as offline benchmarks, which are primarily mobile GUI datasets, but evaluates online performance on *OSWorld*, a desktop-oriented benchmark. This inconsistency raises questions about the comparability of the results.

- The authors should clarify why *OSWorld* (a computer-use benchmark) was chosen over other mobile GUI online benchmarks such as *AndroidWorld*. Alternatively, the paper should include additional results for mobile GUI online benchmarks to maintain consistency across evaluation settings.
- For the Chinese mobile app experiments (86 selected cases), the description is insufficient. The paper should clarify:
  1. Whether these tests are reproducible and independent of device type, login account, or testing time.
  2. How the tasks were constructed, which apps were included, how success or failure was verified, and whether any case studies can illustrate the dataset’s coverage and diversity.

3. Insufficient ablation study
The paper mentions two SFT stages—Stage 1: Perception Task and Stage 2: Planning Task—and reports only final evaluation results. It is recommended to provide separate ablation results showing the performance of each SFT stage on the corresponding evaluation benchmarks to better assess their individual contributions.

**Questions:**

The questions are already included within the weaknesses section.

---

### Official Review · Reviewer_sSYJ · 2025-10-24

**Soundness:** 1
**Presentation:** 2
**Contribution:** 2
**Rating:** 2
**Confidence:** 4

**Summary:**

The paper presents UItron, an open-source foundational model for automated GUI agents that operate across PC and mobile platforms. It integrates systematic data engineering, an interactive infrastructure, and a curriculum reinforcement learning framework.
UItron is trained through supervised fine-tuning on perception and planning tasks, followed by offline and online reinforcement learning. Experiments show strong performance across multiple benchmarks.

**Strengths:**

- Comprehensive design: Combines perception, grounding, and planning within a unified framework.

- Innovative data pipeline: Effective use of backtracking, trajectory distillation, and multimodal data unification.

- Strong empirical results: Outperforms or matches leading models like UI-TARS across diverse benchmarks.

**Weaknesses:**

- The paper lacks clear novelty. Most of its contributions lie in data engineering, while there is little innovation on the algorithmic side. However, many details of the data construction and processing are not disclosed.

- The experimental evaluation is also insufficient. There are too few baselines—models such as Claude 4, OpenAI o3, and UI-TARS-2 should have been included for comparison.

- Moreover, the experiments lack deeper analysis, such as tracking changes in reward, entropy, or other reinforcement learning metrics over training steps. The impact of each strategy (e.g., backtracking) is not clearly isolated or analyzed.

- Finally, the paper omits important implementation details, such as the parameter settings in Equations (5) and (6), and the advantage calculation method in RL stage.

**Questions:**

- For the Perception Data
    - it is unclear which open-source datasets were specifically used.
    - How are the OCR, VQA, and Grounding datasets merged into a GUI-related dataset? What is the format of this GUI-related dataset?

- For the Planning Data, how is it collected?

- For the Distillation Data, the paper mentions *“using single-attempt successes for supervised fine-tuning and multiple-attempt cases for GRPO training.”* However, isn’t this essentially data construction? In GRPO, shouldn’t the model sample trajectories by itself during training rather than relying on pre-collected data?

- For GUI tasks, how is voting performed?

- What is the reward model used in reinforcement learning? Line 320 refers to “multiple scoring models”, could the authors clarify what these models are and how they are combined?

- In offline RL, how is the golden action determined?

- What exactly is the curriculum in the proposed Curriculum Reinforcement Learning (CuRL) framework?

---

### Official Review · Reviewer_bSae · 2025-10-31

**Soundness:** 2
**Presentation:** 3
**Contribution:** 2
**Rating:** 2
**Confidence:** 4

**Summary:**

This paper introduces Ultron, an open-source foundational model for automated GUI agents designed to operate on both PC and mobile devices. The authors develop a comprehensive data pipeline that includes unifying and augmenting perception data, enhancing planning data, and an automated trajectory distillation pipeline using GPT-4O and a VLM-based voting mechanism . An interactive environment connecting PC and mobile devices was built to support both data collection and online reinforcement learning. Moreover, a significant contribution of this work is a new, large-scale dataset of over one million manually annotated operation steps from the top 100 most popular Chinese mobile apps.

**Strengths:**

- **Comprehensive 3-Stage Training Paradigm:** The paper's core strength is its well-conceived 3-stage training strategy. It logically builds the agent's capabilities: first, learning to *see* (Perception SFT) , then learning to *plan* (Planning SFT) , and finally, learning to *explore and reason* in a dynamic environment (CuRL). The ablation study provides strong quantitative support for this curriculum.
- **Major Dataset Contribution:** The paper's most significant and lasting contribution is likely the new Chinese mobile app dataset. Manually annotating over 1 million operation steps across the top 100 apps is a massive undertaking. This dataset and its associated benchmarks (Tables 5 & 6) address a clear and important gap in the literature, enabling more diverse and robust agent development. The outstanding performance of Ultron on this benchmark highlights the value of this in-domain data.
- **Robust and Extensive Evaluation:** The authors demonstrate the robustness of Ultron through a rigorous evaluation across a wide variety of tasks and platforms. This includes GUI perception (VWB, RefExp, etc.) , element grounding (ScreenSpot V1/V2), offline planning (AndroidControl, GUI-Odyssey) , and online planning (OSWorld). Achieving competitive or SOTA results across so many different benchmarks (e.g., highest grounding/SR on AndroidControl , SOTA for GUI agents on OSWorld ) is a strong testament to the quality of the model.
- **Systematic Data Engineering:** The paper provides a clear blueprint for data engineering in the context of GUI agents (Figure 1). The automated trajectory distillation pipeline, which uses GPT-4O for task generation and a VLM-voting mechanism for quality control, is a valuable and scalable approach. The explicit inclusion of "backtracking" data is also an interesting technique for improving the model's state representation.

**Weaknesses:**

- **Missing Model Architecture Details:** The most significant weakness is the complete omission of the base model architecture. The paper does not state which Large Language Model is used for Ultron-7B and Ultron-72B. If both of them are trained from scratch, the authors also need to detailedly introduce the backbone.
- **Vague Details on "Backtracking" Inference:** The concept of "backtracking" is introduced as a key component of Stage 2 SFT . Equation (2)  defines it as a training task. However, the paper never explains *how* this training translates to improved inference. Does the model use this capability during its planning phase (e.g., explicit reflection or self-correction)? Or is it purely a data augmentation strategy to force the model to learn a better latent representation of state transitions? This needs clarification.
- **Ambiguity in General Multimodal Data:** The paper mentions enhancing training with "general multimodal data"  in addition to the GUI-specific data. It briefly lists "image-text pairs from OCR, VQA, and image captioning tasks". To properly assess the contribution of the GUI-specific data, it is crucial to quantify which general-purpose datasets were used and in what proportion they were mixed with the specialized data during Stage 1 SFT.
- **Lack of Ablation Studies:** The paper only introduces limited information regarding the ablation in the three-stage training framework. Could you please provide more details regarding the ablation for Perception Task and Planning Task respectively?

**Questions:**

- **Model Architecture:** Could you please specify the base vision-language model (VLM) architecture used for Ultron-7B and Ultron-72B?
- **Use of Backtracking:** The "backtracking" training task  is an interesting addition. How is this capability leveraged during *inference*? Is it purely a training-time objective to improve state understanding, or does the Ultron agent perform any explicit reflection or correction using this mechanism when it encounters an error?
- **VLM Voting Mechanism:** The VLM-based voting is used for both distillation and online RL rewards. The paper mentions using "multiple scoring models". Which specific models were used in this voting ensemble? Was there any analysis on the inter-annotator agreement (i.e., agreement rate) between these models?
- **Dataset Release:** Can you please confirm that the new, manually annotated 1-million-step dataset for Chinese mobile apps will be made publicly available alongside the model and code?

---

### Official Review · Reviewer_Linm · 2025-11-01

**Soundness:** 3
**Presentation:** 3
**Contribution:** 3
**Rating:** 6
**Confidence:** 3

**Summary:**

UItron is an open-source foundational GUI agent designed for automated operation across PC and mobile environments. It integrates advanced GUI perception, grounding, and planning through a three-stage training paradigm: two supervised fine-tuning stages on perception and planning tasks, followed by a curriculum reinforcement learning stage using the GRPO algorithm. A large-scale interactive infrastructure and data engineering pipeline enable over one million operation trajectories. Experimental results show that UItron-72B achieves state-of-the-art performance on GUI perception and planning benchmarks, particularly excelling in Chinese App scenarios, demonstrating strong real-world generalization and scalability.

**Strengths:**

Originality:
- This integration of multi-stage training and data engineering represents a novel and comprehensive approach to GUI agent development.

Quality:
- The paper demonstrates strong systematic data engineering investigation, multi-model reward validation, and extensive benchmarking.
- The experiments are coherent with the methods, demonstrating the performance of each training stage.

Clarity:
- The paper is clear and well-organized. It elaborates on data engineering and training stages.
- Figures and structured task formulations effectively illustrate the model’s reasoning and interaction process.

Significance:

- UItron establishes a scalable foundation for practical GUI automation, bridging perception and planning within real-world PC and mobile applications.
- Its superior performance in the integration of mobile and PC environments, multilingual tasks, and online/offline scenarios marks a substantial step toward deployable, general-purpose GUI agents.

**Weaknesses:**

- Although the paper contributes to unified interactive infrastructures and data engineering, most methods used in this framework are existing methods (e.g., SFT, GRPO), which might limit its novelty.

- Although UItron employs “backtracking” and structured reasoning formats, its internal decision process remains opaque. Visualization or case studies could reveal whether backtracking contributes to actual causal reasoning.

**Questions:**

- Could you provide some concrete examples of the planning tasks? How do you design the planning data and evaluate planning and backtracking in detail?

---

### Official Review · Reviewer_GN7m · 2025-11-01

**Soundness:** 3
**Presentation:** 3
**Contribution:** 3
**Rating:** 4
**Confidence:** 3

**Summary:**

This paper presents UItron, an open-source foundation model for GUI agents. In the development of UItron, the authors highlight the systematic data engineering, unified interactive infrastructure, and curriculum reinforcement learning framework and their effectiveness for the training of the model. Through extensive experiments on various GUI benchmarks, the paper demonstrates the effectiveness of UItron especially in Chinese mobile app scenarios.

**Strengths:**

1. This paper presents thorough research and engineering efforts in data engineering and unified interactive infrastructure, and demonstrates their effectiveness in developing GUI agents/models. These empirical exploration and findings are potentially valuable for the future development of GUI models.
2. The open-sourced UItron models are beneficial to both research community and the industry as the fundation model for subsequent research and development of GUI agents.

**Weaknesses:**

1. While the paper claims that UItron demonstrates superior performance on perceptual and grounding tasks, judging from the results of UItron-7B vs. UI-TARS-7B in table 2 and 3, it seems to suggest that UItron only demonstrates comparable or slightly worse performance compared with UI-TARS at 7B scale.
2. One of the main contribution of UItron is its leading performance in Chinese mobile app scenarios, however, given that it has put a lot of data engineering efforts specifically into Chinese mobile app, I would not be surprised by the results/findings of their better performance compared with other models that are not specifically tuned for Chinese mobile app.

**Questions:**

Please see the weaknesses above.

---

### Note · Authors · 2025-11-24

I have read and agree with the venue's withdrawal policy on behalf of myself and my co-authors.